# ♚ FACTCHECKMATE: PREEMPTIVELY DETECTING AND MITIGATING HALLUCINATIONS IN LMS

## ABSTRACT

Language models (LMs) hallucinate. We inquire: Can we detect and mitigate hallucinations *before* they happen? This work answers this research question in the positive, by showing that the internal representations of LMs provide rich signals that can be used for this purpose. We introduce FACTCHECKMATE, which preemptively detects hallucinations by learning a classifier that predicts whether the LM *will* hallucinate, based on the model's hidden states produced over the inputs, before decoding begins. If a hallucination is detected, FACTCHECKMATE then intervenes, by adjusting the LM's hidden states such that the model will produce more factual outputs. FACTCHECKMATE provides fresh insights that the inner workings of LMs can be revealed by their hidden states. Practically, both the detection and mitigation models in FACTCHECKMATE are lightweight, adding little inference overhead; FACTCHECKMATE proves a more efficient approach for mitigating hallucinations compared to many post-hoc alternatives. We evaluate FACTCHECKMATE over LMs of different scales and model families (including Llama, Mistral, and Gemma), across a variety of QA datasets from different domains. Our results demonstrate the effectiveness of leveraging internal representations for early hallucination detection and mitigation, achieving over 70% preemptive detection accuracy. On average, outputs generated by LMs with intervention are 34.4% more factual compared to those without intervention. The average overhead difference in the inference time introduced by FACTCHECKMATE is around 3.16 seconds.

## 1 INTRODUCTION

Language models (LMs) hallucinate, a phenomenon where they produce nonfactual or even misleading outputs that often appear plausible (Ji et al., 2023a; Bang et al., 2023; Xu et al., 2024; Zhang et al., 2023; Li et al., 2024; Huang et al., 2023; Ye et al., 2023). Extensive efforts have been devoted to mitigating their hallucination issues (Rawte et al., 2023; Zhou et al., 2021). These approaches are mostly *reactive*, addressing hallucinations *after* they occur, and often require resampling new outputs (Li et al., 2023; Manakul et al., 2023), substantially increasing the inference overhead. In addition, these approaches treat the LM as a black box, while relying on external LMs for detecting hallucinations, missing the opportunity to gain deeper insights into the internal workings of these models.

Recent findings by Azaria & Mitchell (2023) and Burns et al. (2022) show that probing the LMs' representaions can effectively determine the factuality of their outputs. Marks & Tegmark (2023) observe that the hidden states produced by the middle layers of LMs over complete statements exhibit linear separability in binary factuality classification tasks. However, these studies have a relatively narrow focus, primarily addressing hallucination detection in a *reactive* manner. A more thorough investigation is needed.

The key hypothesis of this paper is that, the LMs' hidden states reveals valuable information about their internal working mechanisms, and provide signals that can be used to predict whether it is likely to hallucinate *before* it happens. More formally, we propose FACTCHECKMATE to answer the following research question (RQ): *Can we preemptively predict and mitigate hallucinations with LMs' internal representations?* FACTCHECKMATE learns a classifier that, taking the models' hidden states over the inputs, predicts whether the model *is about to* hallucinate. If a hallucination is detected,

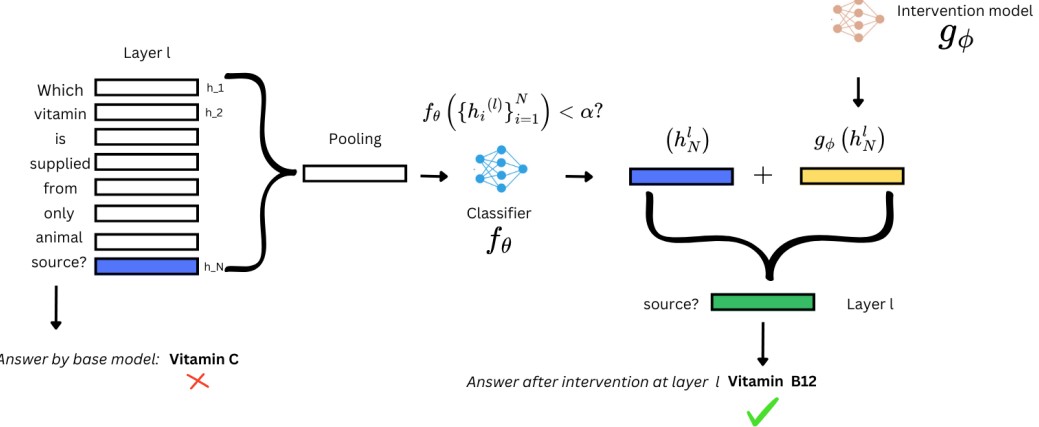

Figure 1: FACTCHECKMATE Pipeline. A demonstration of how preemptive detection and subsequent mitigation work. As shown, at a layer $l$, the hidden states of only the prefix are aggregated and passed to the classifier $f_\theta$. Once hallucination is detected with classification probability $< \alpha$, $\mathbf{g}_\phi$ intervenes and adjusts the last token $\mathbf{h}_N^{(l)}$. This leads to a more factual output than before.

FACTCHECKMATE intervenes, by adjusting the LM's hidden states with a learned invervention model, and steering them towards producing more factual outputs (Figure 1).

Our experiments answer the RQ in the positive. We evaluate FACTCHECKMATE across three QA datasets from domains: NQ-open (Wikipedia; Lee et al., 2019), MMLU (STEM exam; Hendrycks et al., 2020), and MedMCQA (medical; Pal et al., 2022). For all, FACTCHECKMATE successfully predicts whether or not the LMs will hallucinate over 70% of the time, significantly outperforming a 50% random baseline. This is achieved when the LMs have only seen the input questions and before decoding starts. We observe consistent trends across LMs of different scales and familities, including Llama2 (7B and 13B; Touvron et al., 2023a), Llama3/3.1-8B (Dubey et al., 2024) Mistral-7B (Jiang et al., 2023), and Gemma-7B (Team et al., 2024). Furthermore, FACTCHECKMATE's intervention model can effectively improve the LMs' outputs. Using GPT-4o as a judge, which shows high agreement with human evaluations in our experiments, we find that on average, outputs generated by LMs with intervention are 34.4% more factual than those produced without intervention. We also calculate the overhead in the inference time introduced by FACTCHECKMATE, with an average increase of approximately 3.16 seconds, showing minimal impact on inference performance.

FACTCHECKMATE reveals surprising insights into existing LMs, and can potentially lead to more profound understanding of their internal working. All code, data, and checkpoints for reproducing our findings will be released.

We start by presenting the FACTCHECKMATE's hallucination detection model and results in §2, followed by the intervention model and results in §3. Additional experiments and analysis are presented in §4.

## 2 FACTCHECKMATE HALLUCINATION DETECTION

This section focuses on FACTCHECKMATE's preemptive hallucination classifier (§2.1) and experimental results (§2.2).

### 2.1 PREEMPTIVE HALLUCINATION DETECTION WITH A LIGHTWEIGHT CLASSIFIER OVER HIDDEN STATES

**Classifier.** FACTCHECKMATE learns a binary classifier $f_\theta$ to preemptively detect hallucinations. Parameterized by a learned two-layer ReLU-MLP followed by a sigmoid function, $f_\theta$ takes as input the LM's hidden states and outputs the probability that the LM *will* hallucinate. More specifically,

let $\{\boldsymbol{h}_i^{(l)}\}_{i=1}^N$ be a sequence of $N$ hidden states that the LM produces over the input of length $N$. A $d$-dimensional vector $\boldsymbol{h}_i^{(l)}$ denotes the output of the feedforward network (FFN) of the $l$-th transformer layer, at the $i$-th token. The classifier $f_\theta$ takes as input the average over $\{\boldsymbol{h}_i^{(l)}\}$ and produces a scalar between 0 and 1 indicating the probability that the LM will hallucinate in its response to the input:

$$f_\theta\left(\{\boldsymbol{h}_i^{(l)}\}_{i=1}^N\right) = \sigma\left(\text{ReLU-MLP}\left(\frac{1}{N}\sum_{i=1}^N \boldsymbol{h}_i^{(l)}\right)\right) \tag{1}$$

$l$ is empirically determined based on validation performance, and can vary by the LMs and datasets. In general, $l$ tends to be the middle layers. More details about the best empirical layer for each LM can be found in Appendix B.1

We train a separate classifier tailored to each LM.[1] We consider LMs from different families of different scales, including Llama2 (7B and 13B; Touvron et al., 2023a), Llama3/3.1-8B (Dubey et al., 2024) Mistral-7B (Jiang et al., 2023), and Gemma-7B (Team et al., 2024).

**Data collection.**  In order to train $f_\theta$, we need to collect a LM's hidden states over the inputs, and the corresponding binary label indicating whether the LM will produce factual outputs. We construct the training data on three datasets from different domains: NQ-open (Wikipedia; Lee et al., 2019), MMLU (STEM; Hendrycks et al., 2020), and MedMCQA (medical entrance exam; Pal et al., 2022). NQ-open is a QA dataset and contains question and answer pairs. MMLU and MedMCQA are multiple choice datasets, pairing each question with multiple options. We convert MMLU and MedMCQA into a QA dataset by pairing each input question with the gold answer.

To collect the training data for LM $M$, we prompt $M$ with few-shot demonstrations followed by a question, and then collect its hidden states over the inputs. $M$'s output answers are checked against gold ones with the exact match (EM), following standard practice (Gao et al., 2023). If the model's output is wrong, we consider its associated hidden states *will* lead to a hallucination, and vice versa. After producing hidden state and label pairs, we subsample the data to obtain balanced training data containing roughly the same amount of positive (will not hallucinate) and negative (will hallucinate) pairs. In order to compare across different LMs, we create a shared test split across all LMs. Each LM have different training/validation splits. Table 1 summarizes the statistics of the datasets.

$f_\theta$ is trained with a cross-entropy loss on hidden state and label pairs. Early stopping based on the validation accuracy is used.

| Dataset | Total Size | Train (70%) | Validation (15%) | Test (15%) |
|---|---|---|---|---|
| NQ-Open (Lee et al., 2019) | 12,000 | 8,400 | 1,800 | 1,800 |
| MMLU (Hendrycks et al., 2020) | 3,182 | 2,228 | 477 | 477 |
| MedMCQA (Pal et al., 2022) | 3,953 | 2,767 | 593 | 593 |

Table 1: Dataset splits and sizes for training the hallucination classifier $f_\theta$ over the LMs' hidden states (§2.1).

## 2.2 RESULTS

Table 2 shows the hallucination detection test accuracy results. We evaluate the hallucination detection performance using the same classifier $f_\theta$ on different inputs. **I** indicates our *preemptive* classifier, that takes the LMs' hidden states produced over the **input questions only**. **I+O** indicates a *reactive* baseline, which sees the hidden states produced over **a concatenation of the input questions and the LMs' output answers.** It is, therefore, expected that I+O achieves better performance, as it has access to more information.

Throughout all **I** settings across all LMs and datasets, $f_\theta$ achieves well above the 50% random guess baselines. This confirms that LMs' hidden states provide useful signal for predicting their hallucinations preemptively.

---

[1]Our preliminary experiments show that the hallucination classifier underperforms when applied to hidden states produced by a model different than that it is trained for.

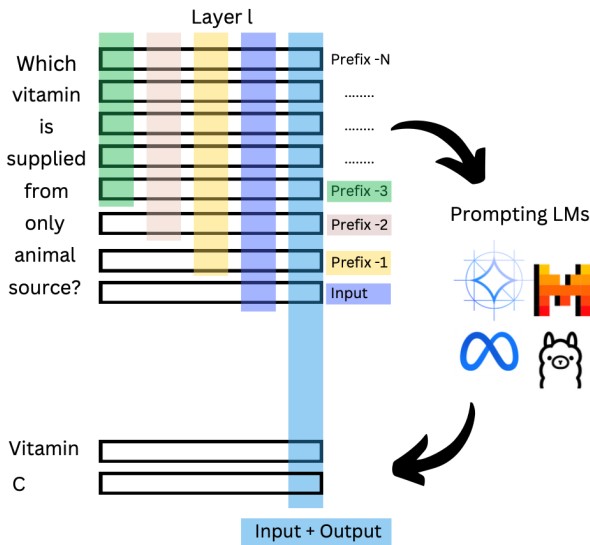

Figure 2: An illustration of different settings used in the experiment. **(Input + Output)** are the hidden states of both the input and output. The subsequent hidden states in the experiment are taken by using only the input or by dropping the last $n$ tokens from the prefix.

| LM | NQ | | | | | MMLU | | | | | MedMCQA | | | | |
|---|---|---|---|---|---|---|---|---|---|---|---|---|---|---|---|
| | | | **Prefix** | | | | | **Prefix** | | | | | **Prefix** | | |
| | I+O | I | -1 | -2 | -3 | I+O | I | -1 | -2 | -3 | I+O | I | -1 | -2 | -3 |
| Llama2-7B | 78.2 | 70.0 | 64.5 | 64.3 | 65.1 | 80.0 | 66.3 | 66.1 | 65.3 | 65.0 | 76.0 | 70.9 | 68.3 | 67.8 | 70.1 |
| Llama2-13B | 81.6 | 76.4 | 74.3 | 74.5 | 73.3 | 84.0 | 68.8 | 69.0 | 68.8 | 68.8 | 77.5 | 70.8 | 68.3 | 67.8 | 70.1 |
| Llama3-8B | 79.4 | 75.9 | 73.4 | 72.2 | 71.6 | 79.0 | 71.1 | 70.8 | 70.3 | 70.3 | 70.8 | 73.0 | 71.8 | 70.7 | 70.7 |
| Llama3.1-8B | 79.2 | 74.9 | 69.1 | 67.7 | 67.6 | 82.5 | 71.5 | 70.5 | 71.3 | 69.5 | 67.5 | 72.9 | 72.3 | 70.0 | 65.8 |
| Mistral-7B | 80.2 | 76.7 | 75.7 | 75.2 | 75.8 | - | - | - | - | - | 69.8 | 70.4 | 69.4 | 69.0 | 69.4 |
| Gemma-7B | 80.2 | 74.5 | 74.4 | 74.2 | 73.9 | 78.1 | 68.8 | 67.2 | 67.2 | 66.1 | 74.7 | 71.6 | 69.5 | 67.9 | 66.8 |

Table 2: Hallucination detection test accuracy. **I+O** indicates a "reactive" baseline that classifies the LMs' hidden states produced over both input questions and output answers, while **I** preemptively classifies hallucinations based on the hidden states over only the inputs. A prefix of $-n$ indicates that the classifier only sees a prefix of the input dropping the last $n$ tokens.

We further make the task more challenging for $f_\theta$, by feeding it with a prefix of the input questions. The results are summarized in the **Prefix** columns. Here, $-n$ indicates that $f_\theta$ sees the hidden states produced over a prefix not including the last $n$ tokens. Illustration of the different input settings for hallucination classification is shown in Figure 2.

In some cases, using a prefix of $-n$ underperforms **I**, while for others their performance is comparable. These results suggest that $f_\theta$ can often predict whether the LM is likely to hallucinate before it even finishes processing the input questions.

On MMLU, Mistral-7B behaves differently than others, and we are not able to produce a sufficiently large test split that is shared between it and others. Therefore, these results are excluded.

# 3 FACTCHECKMATE PREEMPTIVE HALLUCINATION MITIGATION

This section focuses on using FACTCHECKMATE to preemptively mitigate hallucinations, including its intervention model (§3.1) and the experimental results (§3.2).

## 3.1 TRAINING AN INTERVENTION MODEL

When $f_\theta$ detects that LM $M$ is about to hallucinate, FACTCHECKMATE relies on an **intervention model** $\mathbf{g}_\phi$ to mitigate hallucinations preemptively. Conditioning on $\boldsymbol{h}_N^{(l)}$, the LM's last hidden state over the input, $\mathbf{g}_\phi$ generates a $d$-dimensional vector and adds it to $\boldsymbol{h}_N^{(l)}$, before the LM generates any output.

$$\widetilde{\boldsymbol{h}}_N^{(l)} = \boldsymbol{h}_N^{(l)} + \boldsymbol{g}_\phi\left(\boldsymbol{h}_N^{(l)}\right) \tag{2}$$

$\widetilde{\boldsymbol{h}}_N^{(l)}$ is then used in place of $\boldsymbol{h}_N^{(l)}$ for onward LM decoding. The intervention is applied at the last hidden state of the input, as it aligns with the natural progression of decoding and targets the point where hallucinations are most likely to arise.

Intuitively, $\mathbf{g}_\phi$ is supposed to steer the LM's hidden state towards a "target hidden state" $\boldsymbol{h}^{*(l)}$, which is more likely to lead to a factual output. When the LM answers the question correctly, no further modification is needed and $\boldsymbol{h}^{*(l)} = \boldsymbol{h}_N^{(l)}$. When the model answers the question incorrectly, we set the $\boldsymbol{h}^{*(l)}$ to the model's final hidden state *over the input prompt followed by the gold answer.* These target hidden states are paired with their corresponding inputs $\boldsymbol{h}_N^{(l)}$ to train $\mathbf{g}_\phi$. We explore both a deterministic and a stochastic $\mathbf{g}_\phi$:

- **The deterministic $\mathbf{g}_\phi$** is a three-layer ReLU-MLP. It trains by minimizing the mean squared error (MSE) between the adjusted hidden state $\widetilde{\boldsymbol{h}}_N^{(l)}$ and the target one $\boldsymbol{h}^{*(l)}$.

- **The stochastic $\mathbf{g}_\phi$** treats the adjustment vector as a random variable of multivariate Gaussian. It applies a reparameterization trick: $\boldsymbol{g}_\phi(\boldsymbol{h}_N^{(l)}) = \boldsymbol{\mu}(\boldsymbol{h}_N^{(l)}) + \boldsymbol{\epsilon} \odot \boldsymbol{\sigma}(\boldsymbol{h}_N^{(l)})$ for training. Two three-layer ReLU-MLPs are used to for $\boldsymbol{\mu}$ and $\boldsymbol{\sigma}$, with the first two layer shared. Its training objective remains the same MSE loss. One benefit of the stochastic $\mathbf{g}_\phi$ is allowing for sampling the adjustment vectors during inference, which we explore in the experiments.

## 3.2 RESULTS

Figure 3 summarizes the performance of FACTCHECKMATE's intervention model on on the NQ-open dataset, including both the deterministic and stochastic variants. All LMs use the greedy decoding. Following recent works (Raju et al., 2024; Chen et al., 2024b), we employ GPT-4o (OpenAI et al., 2024) as the evaluator to assess for factuality. Human evaluation performed by the authors indicate that there is a substantial agreement between GPT-4o and human judgement, with a Cohen's Kappa of 0.6, justifying our choice of using GPT-4o as an automatic evaluation metric. The specific prompt is provided in Appendix A.

For the stochastic $\mathbf{g}_\phi$, we sample 1, 10, 20, and 30 different $\boldsymbol{\epsilon}$, and apply the interventions with $\mathbf{g}_\phi$; we then use $f_\theta$ to select the intervened hidden state that leads to the highest probability by $f_\theta$, which is then used for onward decoding.[2]

We apply the adjustment only to the first decoding step, modifying $\boldsymbol{h}_N^{(l)}$ to $\widetilde{\boldsymbol{h}}_N^{(l)}$ when the classifier's confidence $\alpha$ is less than or equal to 0.3. As shown in Figure 3, the intervened LMs consistently outperform the base LMs, with a higher proportion of wins favoring the adjusted outputs, with the results varying depending on LM's architecture. The deterministic intervention consistently achieves a win rate of at least 60% in all cases, while without interventions (Base), the LMs show significantly lower performance, with wins as low as 34%. On average, the winning rate of LMs with intervention across all models is 34.4% higher than that of the base LMs.

The results demonstrate that both deterministic and stochastic intervention models improve the factuality of LM's outputs. These finding suggest that, we can mitigate the hallucination even before it shows up in the generation of the language model.

---

[2]A higher probability by $f_\theta$ indicates the hidden state is more likely to lead to a factual output.

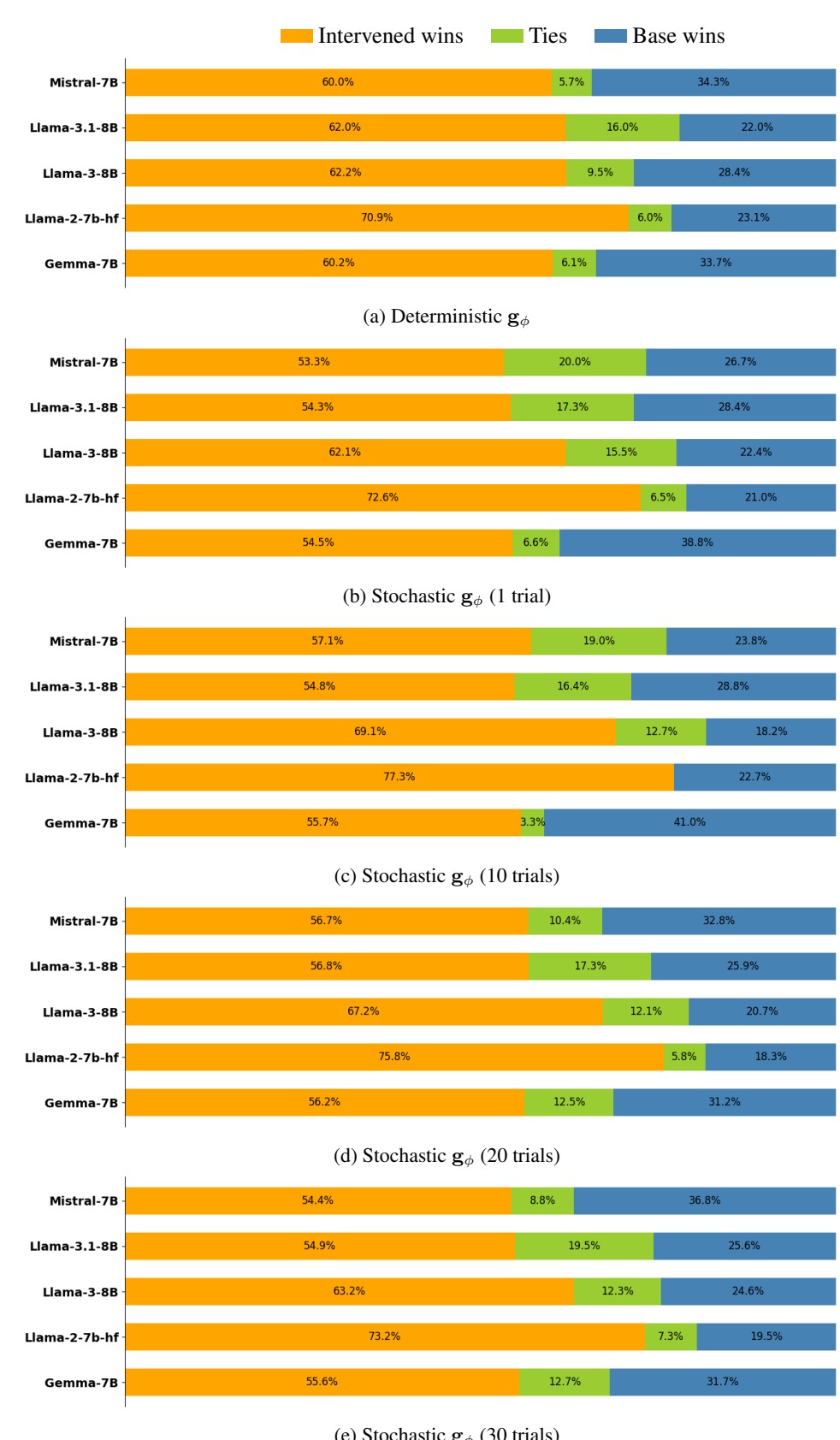

Figure 3: Comparison of FACTCHECKMATE's intervention models. The stochastic model resamples $\epsilon$ for 1, 10, 20, and 30 times, $f_\theta$ used to select the intervened hidden state that leads to the highest probability by $f_\theta$. Green color indicates tie, orange for the intervened LM, and blue for the base LM. (§3.2).

We further compare to Duan et al. (2024), a baseline that applies a Principal Component Analysis (PCA) based approach to engineer the hidden states to mitigate hallucinations in LMs, we refer to this baseline as *PCA*. Figure 4 compares the baseline methods, including *PCA* and a sampling-based decoding approach, which utilizes the hallucination classifier component of FACTCHECKMATE (referred to as Sample-FACTCHECKMATE-CLS).

For *PCA*, adjustments are applied at every generation step, following Duan et al. (2024) with greedy decoding and evaluated against greedy-based decoding version of the non-intervened LM, this to eliminate any effects that sampling might cause. Sample-FACTCHECKMATE-CLS is evaluated against sample-based decoding version of the non-intervened LM, with the same random seed maintained for consistent comparison. As shown, for most models, both baselines result in lower intervened win rates and constantly higher base wins, compared to FACTCHECKMATE in Figure 3.

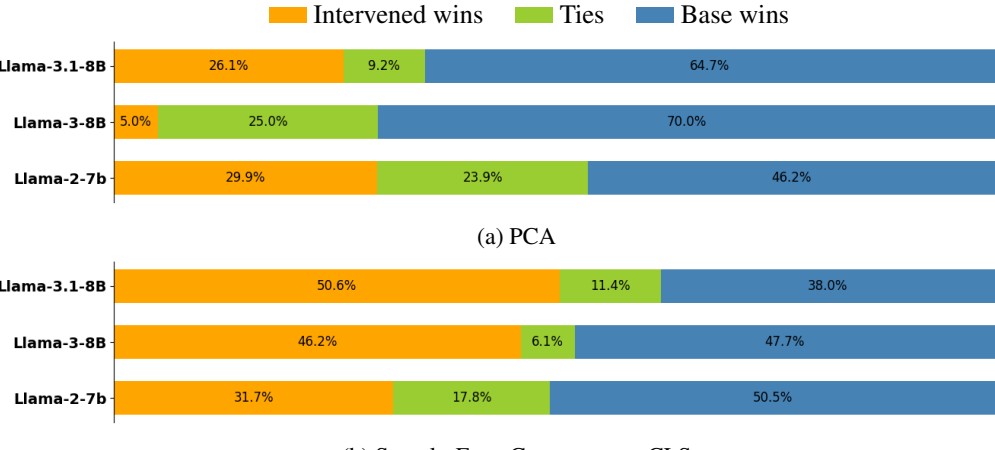

(a) PCA

(b) Sample-FACTCHECKMATE-CLS

Figure 4: Baseline Comparison: The figure shows the winning rate of the intervened LM (Orange), the base LM (Blue), and ties across two different baselines (Green) (§3.2).

## 4 ADDITIONAL EXPERIMENTS

In the following section, we first evaluate the inference time overhead (§4.1). Next, we conduct classification experiments to analyze performance across different modes of aggregation (§4.3) and investigate the role of word embedding layers (§4.2).

### 4.1 EVALUATING FACTCHECKMATE TIME OVERHEADS

Both $f_\theta$ and $\mathbf{g}_\phi$ are lightweight and should incur minimal inference overhead. We confirm this across three models: Llama-2-7B, Llama-3-8B, and Llama-3.1-8B. For each model, the average inference time was measured both with and without FACTCHECKMATE over three runs, each processing 400 few-shot prompts.

| LM | LM Intervention State | Average Inference Time (s) | $\Delta$ (s) |
|---|---|---|---|
| Llama-2-7B | Base | 235.69 | - |
| Llama-2-7B | FACTCHECKMATE | 237.91 | +2.22 |
| Llama3-8B | Base | 272.02 | - |
| Llama3-8B | FACTCHECKMATE | 276.67 | +4.65 |
| Llama3.1-8B | Base | 272.84 | - |
| Llama3.1-8B | FACTCHECKMATE | 275.46 | +2.62 |

Table 3: Comparison of LMs inference time overheads over three runs per LM. The average difference in inference time is approximately 3.16 seconds, showing minimal impact on inference performance.

|  | | | Prefix | | |
|---|---|---|---|---|---|
| LM | I+O | I | -1 | -2 | -3 |
| Llama-2-7b | 63.9 | 52.3 | 55.3 | 54.9 | 55.6 |

Table 4: Results for the word embedding layer of Llama-2-7b on MedMCQA dataset. (§4.2). The figure shows classification accuracy of approximately 50%, indicating no influence of the question difficulty or type on the preemptive hallucination results shown in Table 2.

|  | Mean Pooling | | | | | Last token | | | | | Max pooling | | | | |
|---|---|---|---|---|---|---|---|---|---|---|---|---|---|---|---|
|  | | | Prefix | | | | | Prefix | | | | | Prefix | | |
| LM | I+O | I | -1 | -2 | -3 | I+O | I | -1 | -2 | -3 | I+O | I | -1 | -2 | -3 |
| Llama3-8B | 79.4 | **75.9** | 73.4 | 72.2 | 71.6 | 81.7 | 71.0 | 63.3 | 51.8 | 51.3 | 73.1 | 70.5 | 69.9 | 68.8 | 68.9 |

Table 5: Comparison of hallucination classification across different modes of aggregation for the same layer and LM. Here we show the results for the Llama3-8B on layer 15. We see that the difference between **I+O** and **I** is the least when the mean is the mode of aggregation.

The results are summarized in Table 3. The table shows that FACTCHECKMATE introduces a negligible overhead to the inference process, preserving performance close to that of the baseline. We see that the result is consistent over models. This negligible overhead is a promising factor for scaling the experiments or integrating it into the existing LMs' pipelines.

### 4.2 $f_\theta$ CLASSIFIES THE HIDDEN STATES RATHER THAN THE QUESTIONS

One possible explanation for $f_\theta$'s strong preemptive hallucination detection performance is that it might be classifying the input questions rather than the LMs' hidden states. It is true that more difficult questions could lead to a higher chances of hallucinations by the LMs. However, our results indicate that it is the LMs' hidden states, rather than the questions themselves, that drive the success of $f_\theta$.

Table 4 summarizes the test accuracies for an $f_\theta$ trained and tested on the word embedding layer of Llama-2-7B, before any contextualization by the LM. Across the board, the accuracies are close to 50% random guess. This confirms that, the model is not skewed towards favoring a certain type of question over another while doing the classification. The difficulty of the question is hence, not a contributing factor to the accuracy calculated by classifying the hidden states.

### 4.3 PREEMPTIVE HALLUCINATION DETECTION ACROSS VARIOUS MODES OF AGGREGATION

We explore three modes for aggregating the hidden states before passing them to the classifier: mean pooling, max pooling, and taking the last token. We see that the mean pooling gives us the best accuracy as shown in Fig 5a. To test how different modes of aggregation work for the preemptive experiments, we compare all the three modes. This is done across the same layer for a the same model. As shown in Table 5, we see that the accuracy of the entire sentence (**I+O**) is similar for last token and mean pooling. However, the drop in the subsequent accuracies is the maximum when last token is used. The maximum accuracy for **I** is when mean pooling is used. Therefore, we use mean pooling as our mode of aggregation in all our experiments.

## 5 RELATED WORK

**Definitions.** In this work, we investigate the phenomenon of hallucinations in language models that generate responses based solely on their parametric knowledge, similar to Azaria & Mitchell (2023). This contrasts with in-context generation scenarios where external knowledge sources are explicitly incorporated within the prompt. We adopt the refined taxonomy proposed by Huang et al. (2023), categorizing hallucinations into Factuality or Faithfulness. FACTCHECKMATE focuses its study on addressing Factuality hallucinations, which are further divided into factual inconsistencies and factual fabrications.

**Hallucination Detection.** Hallucination remains a significant issue undermining the language model's usefulness. Existing research on hallucination detection has primarily focused on post-processing methods applied after the inference process is completed and often utilizing external knowledge sources for verification, as in (Manakul et al., 2023; Li et al., 2023; Chern et al., 2023). For instance, Gou et al. (2024) introduce CRITIC, a framework that validates model outputs through tool interaction, and FACTSCORE proposed by Min et al. (2023), is a fine-grained factual accuracy metric that breaks down generated content into atomic facts, assessing their accuracy by comparing them against reliable sources.

A recent promising line of research leverages the internal mechanics of language models to detect hallucinations. Works such as (Burns et al., 2024; Azaria & Mitchell, 2023; Marks & Tegmark, 2023) are pioneering efforts to assess the truthfulness of outputs by examining the hidden states of language models. The work by Meng et al. (2022) locates where factual associations are stored in GPT models. These studies have spurred further research into using LLMs' internal representations in hallucination detection (Chen et al., 2024a; CH-Wang et al., 2024). For instance, the MIND framework, introduced by Su et al. (2024), generates training data in unsupervised approach for training hidden states based hallucination detectors. Duan et al. (2024) conducts an experimental examination of the hidden states of LLMs when processing factual versus nonfactual responses. Following this line of research, FACTCHECKMATE showcases the effectiveness of preemptive hallucination detection, i.e. identifying warning signals several tokens before the hallucinations actually occur, via solely exploiting the language model's hidden states.

**Hallucination Mitigation.** In the realm of hallucination mitigation at inference time, existing work has explored self-correction and automated feedback approaches, where the language model is prompted to fix its generation flaws, with or without leveraging feedback from the model itself or some external knowledge source, as detailed in (Pan et al., 2023; Dhuliawala et al., 2023; Ji et al., 2023b). A recent approach to mitigating hallucinations involves utilizing activation engineering (Subramani et al., 2022), first applied to hallucination mitigation by Duan et al. (2024). FACTCHECKMATE builds on these findings and explores additional activation engineering techniques to intervene and mitigate hallucinations during inference time.

## 6 CONCLUSION

In conclusion, FACTCHECKMATE demonstrates that the hidden states of language models encode rich information that can be used to predict hallucination preemptively, even before they appear in the generated output. In FACTCHECKMATE, leveraging this insight, we develop an intervention mechanism that steers the LM's generation towards more factual outputs, once the hallucination is detected. We achieve a preemptive hallucination detection accuracy of more than 70%, and an average of 34.4% more factual output by LMs supported by FACTCHECKMATE, compared to the base LMs. FACTCHECKMATE empirically proves the significant potential of utilizing the internal working of LMs, through learning lightweight models for hallucination detection and mitigation, introducing only a negligible average overhead of 3.16 seconds to the inference time.

## 7 LIMITATIONS AND FUTURE WORK

We have only looked at the hidden states as an internal component for classification to predict the factuality of a sentence. Exploring other LM's internal components presents a potential direction for future work. With our detection and intervention experiments, we see that different layers in the LM have varying effects. The classifier $f_\theta$ and the intervention model $g_\phi$, are sensitive to the hyperparameters selected. To solve this, we want find out a more robust and consistent approach that is less sensitive to varying hyperparameters, to steer the generation of the model towards the truth.

Going ahead, we aim to build a pipeline that is more generalizable and is applicable to a variety of domains. Expanding our evaluation to include diverse datasets with different distributions could provide valuable insights and a potential future direction for improving model generalizability. FACTCHECK-MATE has shown promising results in question-answering tasks, and it would be interesting to extend its application to other tasks, including dialogue-based and long-form generation tasks.

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

## A  FACTUAL ASSESSMENT PROMPT

To assess factual accuracy, we use GPT-4o (OpenAI et al., 2024) as the evaluator. To reduce stochasticity in the prompting process, we set the temperature to $1 \times 10^{-14}$ and top_p to $1 \times 10^{-17}$. The prompt used for evaluation is as follows:

> **System:** *You are an expert evaluator with an access to Google Search. Your task is to evaluate two responses to a question for factual accuracy. For this task, 'Factual accuracy' refers to the correctness and relevance of the information, aligned with facts accepted or verified as recent as 2021. Ignore stylistic differences, length, opinions, or phrasing unless they change the factual meaning. Supported by your Google Search results, decide which response, if any, is correct. Answer 'first' if the first response is the only correct response, 'second' if the second response is the only correct response, 'both' if both responses are correct, or 'neither' if neither response is correct or if the information provided is ambiguous or insufficient for making a decision, You should favor the response that shows uncertainty if the other response is incorrect. Then, in a new line, briefly explain the reason.*
> **User**: *Question:* who played first game in world cup 2018? *First Response:* Russia vs Saudi Arabia *Second Response:* Brazil vs Germany.

## B  EXPERIMENTS FOR CLASSIFICATION

### B.1  HIDDEN REPRESENTATION CLASSIFICATION ANALYSIS

Given the datasets and models described above, for every layer in a model we train a corresponding classifier on hidden states of that respective layer. We use three modes for aggregating the hidden states before passing them to the classifier: mean pooling, max pooling and taking the last token in the hidden states. Figure 5b illustrates the accuracy of hallucination detection of the classifiers for the entire sequence, using the mean token representation for aggregation. As shown, the accuracy across all evaluated models mostly exceeds 0.75, indicating a robust capability to identify hallucinations. This high level of performance underscores the efficacy of the hidden state representations in distinguishing factual accuracies within generated content. As seen in the figure, we also see that the accuracy peaks for the middle layers. The best performing layer per model per dataset is shown in Table 6 ur experiments also explore taking the elementwise max over hidden states, or taking the last one as the input to $f_\theta$, and find they slightly underperform taking the average.

Therefore, for all models we calculate the test accuracy across all layers and all modes of aggregation. Quantitative results are shown in the first column of Table 2. Given the three modes of aggregation, we see that mean pooling gives the best results in most cases. Figure 5a shows the test accuracy per layer per mode.

**Setup**: The classifier is trained using an Adam optimizer with a learning rate of $10^{-4}$ with a dropout rate of 0.1. We train all classifiers for 50 epochs.

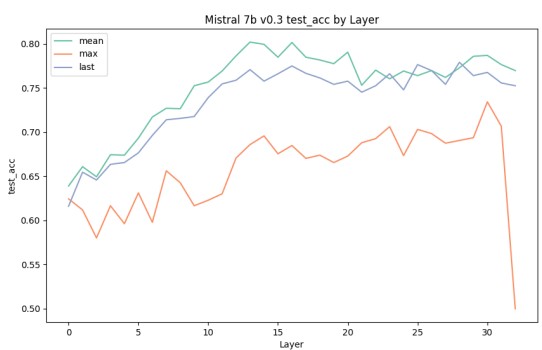

(a) Test Accuracy by Layer for all modes for the model Mistral-7b

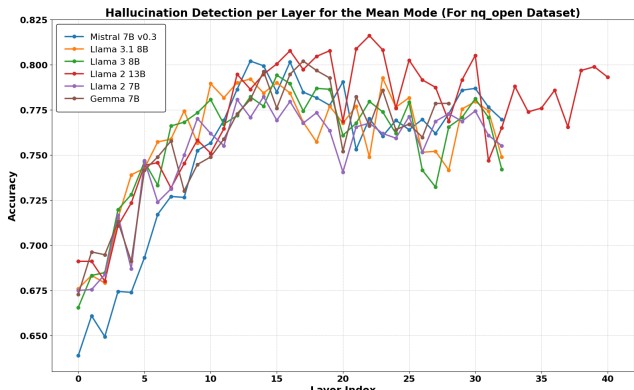

(b) Accuracies for entire sentence across models and layers.

| LM | NQ | MMLU | MedMCQA |
|---|---|---|---|
| Llama-2-7b-hf | 14 | 16 | 14 |
| Llama-2-13b-hf | 22 | 15 | 14 |
| Llama-3-8B | 15 | 17 | 11 |
| Llama3.1-8B | 23 | 14 | 15 |
| Mistral-7B | 13 | - | 12 |
| Gemma-7B | 17 | 17 | 18 |

Table 6: Best Performing layer per model and dataset

