# OpenReview forum: "FactCheckmate: Preemptively Detecting and Mitigating Hallucinations in LMs"
_ICLR.cc/2025/Conference — ICLR 2025 Conference Withdrawn Submission_

### Official Review · Reviewer_asFJ · 2024-10-23

**Soundness:** 2
**Presentation:** 3
**Contribution:** 2
**Rating:** 3
**Confidence:** 5

**Summary:**

This paper presents an approach to detect and mitigate hallucinations in language models (LMs) before they occur. The authors introduce a system named FactCheckmate, which leverages the hidden states of LMs to predict potential hallucinations. This is achieved through a classifier that assesses whether the LM is likely to produce a hallucination based on the internal signals from its hidden states. When a hallucination is detected, FactCheckmate intervenes by adjusting the hidden states to steer the model towards generating more factual content.

**Strengths:**

Overall, the presentation is well-written and easy to follow.

**Weaknesses:**

1. Although many methods for detecting and mitigating LLM hallucinations are outlined in the related work, the authors compare their approach with only one method. To convincingly demonstrate the superiority of their method, it would be prudent to include 3-4 baselines for both detection and mitigation aspects. Without this broader comparison, I cannot recognize the advantages of the authors' approach.

2. I appreciate the experiments conducted on different open-source model families, but there is limited information on the performance of the method on Llama2 Chat and Llama3 Instruct models. Furthermore, it's unclear how the method performs on larger models like Llama2 70B and Llama3 70B. This raises questions about the scalability and generalizability of the proposed approach across various model sizes.

3. While the authors compare their method, FactCheckMate, under random sampling conditions, its effectiveness significantly diminishes from previous levels above 60% to now below 50%. This indicates that FactCheckMate may not be as robust under varied sampling conditions.

4. The claim of a 3.16-second average time in the abstract lacks rigor. Details about the GPU and CPU environments where these measurements were taken are not provided. Additionally, the use of 400 few-shot prompts does not offer a comprehensive view of performance. It would be beneficial to see how the method performs under long-context scenarios to better understand its effectiveness.

5. Regarding the training of an intervention model, I can't find which dataset was used or discuss the hyperparameters involved in detail. More thorough discussion and transparency about the training conditions and parameters would enhance the credibility and reproducibility of the research.

**Questions:**

See the weakness.

---

### Official Review · Reviewer_SFMA · 2024-10-30

**Soundness:** 3
**Presentation:** 3
**Contribution:** 3
**Rating:** 5
**Confidence:** 4

**Summary:**

This paper explores the possibility of preemptively detecting and mitigating hallucinations in language models (LMs). The authors present FACTCHECKMATE, a method that learns to predict whether an LM will hallucinate based on the model's hidden states before decoding begins. If a hallucination is predicted, FACTCHECKMATE adjusts the LM's hidden states to produce more factual outputs. The method leverages the rich signals provided by the internal representations of LMs. Experimental results demonstrate that FACTCHECKMATE achieves over 70% preemptive detection accuracy and increases the factualness of LM outputs by an average of 34.4% compared to no intervention. The inference time overhead introduced by FACTCHECKMATE is approximately 3.16 seconds.

**Strengths:**

**Strength 1** The paper introduces a new approach to detecting hallucinations by leveraging the internal representations of LMs.

**Strength 2** The experimental design is solid, and the results effectively demonstrate the effectiveness of the proposed method.

**Strength 3** The paper is well-written and easy to follow, with clear explanations of the methodology and results.

**Weaknesses:**

**Weakness 1** The paper focuses solely on close-book hallucinations, whereas many hallucinations occur in open-book settings, such as in abstractive summarization. Evaluating the method's effectiveness in handling open-book hallucinations would provide a more comprehensive understanding of its capabilities.

**Weakness 2** The evaluation of the proposed method's factuality is conducted on the NQ-open dataset, and the classifier used is also trained on the same dataset. It remains unclear whether the method can generalize to other datasets, which is crucial for demonstrating the robustness of the approach.

**Weakness 3**: There is no discussion regarding the potential impact of the proposed method on nominal (non-hallucinatory) questions. Since the classifier might have a false positive rate (FPR), it is important to understand how the intervention affects the performance on questions that do not contain hallucinations.

**Questions:**

**Question 1** Could the authors test the method on abstractive summarization tasks to demonstrate whether it performs well in open-book settings? This would help validate the method’s applicability across different types of hallucinations.

**Question 2** Could the authors verify the proposed method’s generality by evaluating its performance on dataset other than NQ-open? Understanding the method’s effectiveness across diverse datasets is essential for demonstrating its robustness.

**Question 3** Could the authors apply the proposed method to benchmarks like Alpaca-eval? It would be interesting to see how the intervention affects the model’s performance on nominal questions and whether there is any degradation in accuracy due to false positives.

---

### Official Review · Reviewer_tLiQ · 2024-11-01

**Soundness:** 2
**Presentation:** 1
**Contribution:** 1
**Rating:** 3
**Confidence:** 5

**Summary:**

This paper introduces FACTCHECKMATE, a framework aimed at preemptively detecting and mitigating hallucinations in language models (LMs) by analyzing their hidden states before generating responses. The approach leverages lightweight classifiers to detect hallucinations early and an intervention model to adjust hidden states for improved factual accuracy.

**Strengths:**

- Extensive experimental results across different models and datasets are robust and demonstrate effectiveness.
- Offers practical implications for real-world applications where factual accuracy is crucial, enhancing LM reliability.

**Weaknesses:**

- Writing needs improvement, including but not limited to the abstract and introduction
- Typos, e.g. Line 44 "representaions"
- determining the factuality through merely probing the LMs' representations is not novel as a methodology
- Limited exploration of other LM components beyond hidden states.
- Generalizability of results is uncertain for tasks beyond QA.

**Questions:**

- Can the authors elaborate on how hyperparameter sensitivity impacts the intervention model’s reliability?
- Is FACTCHECKMATE adaptable to more complex generative tasks, like dialogue or long-form generation?

---

### Official Review · Reviewer_JonF · 2024-11-05

**Soundness:** 2
**Presentation:** 3
**Contribution:** 2
**Rating:** 5
**Confidence:** 4

**Summary:**

This paper introduces a novel approach that predicts and mitigates hallucinations during the generation process by learning the internal representations of language models. This method is forward-looking in that it seeks to intervene before hallucinations occur.

**Strengths:**

1. Both the detection and mitigation models are lightweight, resulting in minimal inference overhead, which is advantageous for practical applications.
2. The approach is evaluated across various large-scale models, including Llama, Mistral, and Gemma, demonstrating its broad applicability.

**Weaknesses:**

1.	The paper lacks an analysis of the generalizability of the learned classification network and intervention model. Specifically, it is unclear whether the trained classification and intervention models are generalizable across different large models and tasks. Given that the data collection for training was based on only three tasks, questions remain regarding the generalizability to other tasks. Is a new training dataset needed for additional tasks, or does the current model extend effectively?
2.	The dataset construction raises some issues or lacks clarity. For certain tasks, it may be straightforward to judge whether the generated output is correct. However, in the case of generative tasks—particularly when the output is lengthy—it becomes challenging to determine whether the output from the large model is accurate, and thereby to ascertain whether the label indicates hallucination. This aspect is not thoroughly addressed in the paper.
3.	For different large models, it is necessary to reconstruct training datasets and train distinct classifiers and intervention networks, making this process relatively complex. Although it may not increase inference time, the time required for dataset construction and model training is also significant and should not be overlooked.
4.	There is a lack of analysis regarding the structure of the classifier and intervention models used. Specifically, the classifier is implemented as a two-layer MLP, and the perturbation model as a three-layer MLP. Details such as the hidden dimensions of these MLPs, and the potential performance impact of adding or reducing layers, are not discussed. Moreover, it is unclear how alternative models, such as transformers, might affect performance.
5.	The paper does not provide specific details on the training setup for the classifier and intervention models, which creates challenges for reproducibility. In my view, training the classification network and intervention model should be central to this method, but there is limited discussion provided.
6.	The experiments presented are insufficient to substantiate the effectiveness of the proposed method. The experimental section primarily compares the base model, but numerous methods already exist for hallucination detection and mitigation in large models, such as PoLLMgraph, SELFCHECKGPT, and TruthX. These more advanced baselines are not included in the comparisons. Additionally, the number of datasets used appears limited, potentially insufficient to demonstrate broad effectiveness across various tasks and datasets. I recommend conducting experiments on a wider range of datasets to strengthen the validation.
7.	Clearly, this method, which relies on internal states, cannot be applied to black-box large models like GPT. This point should be included in the limitations section of the paper.

**Questions:**

See weaknesses.

---

### Note · Authors · 2024-11-26

I have read and agree with the venue's withdrawal policy on behalf of myself and my co-authors.